# Clopidogrel Administration Impairs Post-Stroke Learning and Memory Recovery in Mice

**DOI:** 10.3390/ijms241411706

**Published:** 2023-07-20

**Authors:** Marina Paul, Jonathan W. Paul, Madeleine Hinwood, Rebecca J. Hood, Kristy Martin, Mahmoud Abdolhoseini, Sarah J. Johnson, Michael Pollack, Michael Nilsson, Frederick R. Walker

**Affiliations:** 1School of Biomedical Sciences and Pharmacy, College of Health, Medicine and Wellbeing, University of Newcastle, Callaghan, NSW 2308, Australia; rebecca.hood@adelaide.edu.au (R.J.H.); kristy.martin@newcastle.edu.au (K.M.); 2Hunter Medical Research Institute, 1 Kookaburra Circuit, New Lambton Heights, NSW 2305, Australia; jonathan.paul@newcastle.edu.au (J.W.P.); madeleine.hinwood@newcastle.edu.au (M.H.); michael.pollack@health.nsw.gov.au (M.P.); michael.nilsson@newcastle.edu.au (M.N.); 3Centre for Rehab Innovations, University of Newcastle, Callaghan, NSW 2308, Australia; sarah.johnson@newcastle.edu.au; 4School of Medicine and Public Health, College of Health, Medicine and Wellbeing, University of Newcastle, Callaghan, NSW 2308, Australia; 5Discipline of Anatomy and Pathology, School of Biomedicine, Faculty of Health and Medical Sciences, The University of Adelaide, Adelaide, SA 5005, Australia; 6School of Engineering, College of Engineering, Science and Environment, University of Newcastle, Callaghan, NSW 2308, Australia; mahmoud.abdolhoseini@newcastle.edu.au; 7Department of Clinical Neuroscience, Institute of Neuroscience and Physiology, The Sahlgrenska Academy, University of Gothenburg, 405 30 Gothenburg, Sweden; 8LKC School of Medicine, Nanyang Technological University, Singapore 639798, Singapore

**Keywords:** stroke, microglia, clopidogrel, cortex, neurons, blood vessels, immune cells

## Abstract

Clopidogrel, which is one of the most prescribed antiplatelet medications in the world, is given to stroke survivors for the prevention of secondary cardiovascular events. Clopidogrel exerts its antiplatelet activity via antagonism of the P2Y12 receptor (P2RY12). Although not widely known or considered during the initial clinical trials for clopidogrel, P2RY12 is also expressed on microglia, which are the brain’s immune cells, where the receptor facilitates chemotactic migration toward sites of cellular damage. If microglial P2RY12 is blocked, microglia lose the ability to migrate to damaged sites and carry out essential repair processes. We aimed to investigate whether administering clopidogrel to mice post-stroke was associated with (i) impaired motor skills and cognitive recovery; (ii) physiological changes, such as survival rate and body weight; (iii) changes in the neurovascular unit, including blood vessels, microglia, and neurons; and (iv) changes in immune cells. Photothrombotic stroke (or sham surgery) was induced in adult male mice. From 24 h post-stroke, mice were treated daily for 14 days with either clopidogrel or a control. Cognitive performance (memory and learning) was assessed using a mouse touchscreen platform (paired associated learning task), while motor impairment was assessed using the cylinder task for paw asymmetry. On day 15, the mice were euthanized and their brains were collected for immunohistochemistry analysis. Clopidogrel administration significantly impaired learning and memory recovery, reduced mouse survival rates, and reduced body weight post-stroke. Furthermore, clopidogrel significantly increased vascular leakage, significantly increased the number and appearance of microglia, and significantly reduced the number of T cells within the peri-infarct region post-stroke. These data suggest that clopidogrel hampers cognitive performance post-stroke. This effect is potentially mediated by an increase in vascular permeability post-stroke, providing a pathway for clopidogrel to access the central nervous system, and thus, interfere in repair and recovery processes.

## 1. Introduction

Clopidogrel, which is one of the most prescribed antiplatelet medications in the world, is given to stroke survivors for secondary stroke prevention. Clopidogrel exhibits few known major side effects, which has contributed to its popularity. Since its approval by the US Food and Drug Administration in 1997, clopidogrel has been prescribed to over 90 million patients worldwide [1]. Clopidogrel exerts its antiplatelet activity via antagonism of the P2Y12 receptor (P2RY12) [2]. Peripherally, P2RY12 is primarily located on platelets, where it is an important regulator of platelet activation and aggregation during the blood-clotting process [3]. Although not widely known or considered during clinical trials, P2RY12 is also expressed on microglia, which are the resident immune cells of the central nervous system (CNS) [4,5,6].

Microglia affect a remarkable variety of functions, including regulation of synaptic plasticity associated with learning and memory, responding to CNS damage, and facilitating CNS repair [7]. Highlighting the importance of microglial response after CNS injury, experiments that either conditionally removed or blocked microglial activity after experimental stroke showed increased neuronal loss and infarct size [8,9]. Real-time in vivo imaging studies demonstrated that one of the principal ways that microglia respond to damage is via the extension of cellular processes and cell body migration toward sites of damage [10]. Microglial process extension appears to be chemotactic in nature. One particularly well-characterized microglial chemotactic pathway involves P2RY12 [4,11,12,13]. P2RY12s are exclusively expressed on microglia within the brain [4,6,14] and signaling via P2RY12 is not only required for process extension toward sites of damage but also seems to be involved in microglia recruitment to other sites of interest, such as neuronal dendrites and spines during synaptic plasticity [15,16,17].

Under normal circumstances, neither clopidogrel nor its metabolites are able to cross the blood–brain barrier (BBB). However, structural disruption of endothelial cell junctions of the BBB and increased barrier permeability are pathological characteristics of both ischemic and hemorrhagic stroke in humans [18]. Following a stroke, bloodborne cells, chemicals, and fluid extravasate into brain parenchyma across the impaired BBB as a result of increased paracellular and transcellular permeability [19]. Therefore, clopidogrel and its active metabolites may enter the affected CNS, resulting in the suppression of P2RY12-mediated microglial activation. Indeed, animal studies strongly support this possibility [20,21]. For instance, Lou et al. [21] demonstrated that microglial process movement toward sites of vascular injury was impaired in mice treated with clopidogrel. Similarly, when microglial sensing was blocked in a mouse model of ischemic stroke using a highly potent P2RY12 antagonist, namely, PSB0739, microglia no longer responded to damage, leading to a larger lesion and disrupted functional connectivity [20]. Collectively these findings suggest that clopidogrel may be able to enter the brain after stroke due to an increase in vascular permeability and interfere with normal repair processes facilitated by microglia.

Clinical studies that examined the effect of secondary cardiovascular prevention strategies of vascular risk via antiplatelet regimens on long-term cognitive impairment provided mixed results. The Prevention Regimen for Effectively Avoiding Second Strokes (PRoFESS) trial showed no significant differences in median mini-mental state examination (MMSE) score between those given aspirin and dipyridamole (a non-P2RY12-based mechanism), or aspirin and clopidogrel [22]. The trial time frame was likely too short (the mean duration of follow-up was 2.5 years) to show the effects of the medication on cognitive function, which may unfold over longer timeframes. More recently, a population-based study assessed the relationship between cognitive impairment and vascular secondary prevention up to 10 years after a stroke and found a reduced risk of cognitive impairment following aspirin and dipyridamole combinational treatment but not following clopidogrel monotherapy [23]. These clinical studies indicate that there is still no clear evidence of the effect of secondary prevention strategies of vascular risk on long-term cognitive impairment. Therefore, it is essential to further characterize the potential links between post-stroke treatment with P2RY12 inhibitors and cognitive decline due to their inhibition of microglial P2RY12.

Our study aimed to investigate the impact of clopidogrel on motor and cognitive outcomes in a clinically relevant scenario where clopidogrel was administrated post-stroke. Specifically, we assessed motor function using the cylinder task and associative memory and learning using Bussey–Saksida mouse touchscreen chambers. Touchscreen-based assessments of cognition provide a more translationally relevant and stress-free method for assessing cognition in mice. Moreover, touchscreen testing in mice enables the assessment of specific cognitive domains that are directly relevant to impairments described in human stroke survivors [24]. We further examined the impact of short-term post-stroke clopidogrel administration on physiological parameters, including the survival rate and body mass. Finally, we investigated the impact of the short-term post-stroke clopidogrel administration on changes in the neurovascular unit, with a specific focus on blood vessels, microglia, neurons, and immune cells.

## 2. Results

### 2.1. Short-Term Clopidogrel Administration Impaired Memory and Learning Post-Stroke

Associative memory was assessed using the paired-associate learning (PAL) task. In the PAL task, mice learned that each of three visual stimuli (flower, airplane, spider) was rewarded with a strawberry milkshake only when the correct symbol was displayed in the correct specific location on the screen (left, middle, right). In any given trial of the PAL task, two visual stimuli were displayed on the screen; one image was displayed in its correct location (S+), while the other is displayed in an incorrect location (S−). Specifically, responses to the flower stimulus were correct only when it was located on the left; responses to the airplane stimulus were correct only when it was located in the middle, and responses to the spider stimulus were correct only when it was located on the right. Six different types of trials occurred during the PAL task and mice were required to respond to the S+ to receive a strawberry milkshake, after which they were presented with a new trial. Responses to the S− resulted in the absence of a strawberry milkshake, no tone, a 5 s house light on, then a 20 s time-out period, followed by a correction trial (Figure 1A,B).

Within the first week of post-stroke testing, no statistically significant differences in correct response rates (associated memory) were observed between the sham-operated, stroke + control-treated, and stroke + clopidogrel-treated mice. Within the second week of testing, however, correct response rates for the sham-operated mice (*p* = 0.0098) and stroke + control-treated mice (*p* = 0.0041) were significantly greater than those observed for the stroke + clopidogrel-treated mice. Furthermore, upon comparing the week 2 versus week 1 outcomes, a statistically significant improvement in the correct response rate was observed for both the sham-operated mice (*p* = 0.0002) and stroke + control-treated mice (*p* = 0.0002), but the stroke + clopidogrel-treated mice showed no significant improvement (*p* = 0.1574). Therefore, the sham-operated mice and stroke + control-treated mice demonstrated statistically significant improvements in their associated memory across the 14 days, whereas the stroke + clopidogrel-treated mice did not (Figure 1C).

### 2.2. Short-Term Clopidogrel Administration Did Not Alter Motor Impairment

Pre- and post-stroke, all three groups were evaluated using a spontaneous forelimb asymmetry task. Specifically, this evaluated the paw preference that mice exhibited for stabilizing themselves while rearing within a cylinder (Figure 2A). There were no significant differences in paw preference prior to stroke induction. Differences were noted post-stroke. Specifically, the stroke + control-treated mice exhibited a significantly stronger preference for using their unaffected paw relative to the sham-operated mice on days 2 (*p* = 0.0031), 7 (*p* = 0.0001), and 14 (*p* < 0.0001) post-stroke. Similarly, the stroke + clopidogrel-treated mice exhibited a significantly stronger preference for using their unaffected paw relative to the sham-operated mice at all post-stroke time points (*p* ≤ 0.0001). Additionally, significant differences were observed on days 2 and 14 between the stroke + control-treated mice and stroke + clopidogrel-treated mice (*p* = 0.0437 and *p* = 0.0217, respectively) (Figure 2B).

### 2.3. Short-Term Clopidogrel Administration Decreased Mouse Survival Rates and Body Weight

To assess the impact of clopidogrel on the mice, we measured changes in the survival rates and body weight across the 14 days of clopidogrel administration (Figure 3). Mouse survival rates were recorded daily throughout the experiment. Although all mice were reared in the same environment following a photothrombotic stroke or sham surgery, 3 out of 26 mice from the stroke + clopidogrel treatment died during the 14 days of clopidogrel administration (88.46% survival rate), whereas 0 out of 27 mice from each of the sham-operated (100% survival rate; *p* = 0.0285) and stroke + control-treated mice died (100% survival rate; *p* = 0.0285) (Figure 3A).

Each mouse’s body weight was recorded daily throughout the experiment. Overall, the body weight change was calculated as a percentage (%) change from baseline (day 0) to the last day of clopidogrel administration (day 14). Compared with the sham-operated mice, there was a significant decrease in body mass of both the stroke + control-treated mice (*p* = 0.0016) and stroke + clopidogrel-treated mice (*p* < 0.0001) across the 14 days, but there was no overall significant difference between the stroke + control-treated mice and stroke + clopidogrel-treated mice (*p* = 0.0591) across the 14 days. Looking at the results more closely, on day 1 post-stroke, we recorded a significantly larger decrease in body mass in both the stroke + control-treated mice (*p* < 0.0001) and stroke + clopidogrel-treated mice (*p* < 0.0001) than in the sham-operated mice; however, there was no significant difference in body mass between the stroke + control-treated and stroke + clopidogrel-treated mice (*p* = 0.0693). On day 14 post-stroke, there was no significant difference in body mass between the stroke + control-treated and sham-operated mice (*p* = 0.0895); however, the body mass of the stroke + clopidogrel-treated mice remained significantly decreased compared with the sham-operated mice (*p* = 0.0046) (Figure 3B).

### 2.4. Short-Term Clopidogrel Administration Did Not Alter Vasculature Post-Stroke

Collagen IV labeling was visualized to assess any changes in the vasculature in both the ipsilateral and contralateral cortexes. Within the ipsilateral cortex, there were no significant differences in the total area covered by collagen IV-labeled vessels between the three groups; however, the number of collagen IV-labeled vessels detected was significantly reduced in the stroke + control-treated mice (*p* = 0.0027) and stroke + clopidogrel-treated mice (*p* = 0.0140) compared with the sham-operated mice. There was no difference in the number of collagen IV-labeled vessels between the stroke + control-treated mice and stroke + clopidogrel-treated mice (Figure 4A).

Within the contralateral cortex, there were no significant differences in the total area covered by collagen IV-labeled vessels nor in the number of collagen IV-labeled vessels between the three groups (Figure 4B).

### 2.5. Short-Term Clopidogrel Administration Increased Vascular Leakage Post-Stroke

The IgG staining within the ipsilateral and contralateral cortex regions was visualized to assess stroke-induced cerebrovascular leakage (Figure 5A). Compared with the sham-operated mice, there was a significant increase in the IgG labeling within the ipsilateral cortex region of both the stroke + control-treated mice (*p* = 0.0124) and stroke + clopidogrel-treated mice (*p* < 0.0001). Additionally, the IgG labeling in the stroke + clopidogrel-treated mice was significantly elevated beyond that of the stroke + control-treated mice (*p* = 0.0278) (Figure 5B).

Within the contralateral cortex, there was no significant change in the IgG labeling between the sham-operated mice and stroke + control-treated mice (*p* = 0.1173), nor between the stroke + control-treated mice and stroke + clopidogrel-treated mice (*p* = 0.1566). However, the IgG labeling in the stroke + clopidogrel-treated mice was significantly elevated compared with the sham-operated mice (*p* = 0.0299) (Figure 5C).

### 2.6. Short-Term Clopidogrel Administration Changed the Microglia Number and Morphology Post-Stroke

Iba1 labeling, which is an established microglia marker, was visualized to assess the number and the appearance of microglia within the ipsilateral and contralateral cortexes. Within the ipsilateral cortex, automated microglia detection (using MicroTrac software) revealed that compared with the sham-operated mice, the microglia number was significantly elevated within the ipsilateral cortex of both the stroke + control-treated mice (*p* = 0.0124) and stroke + clopidogrel-treated mice (*p* < 0.0001). Additionally, the microglia number for the stroke + clopidogrel-treated mice was significantly elevated beyond that of the stroke + control-treated mice (*p* = 0.0278). Morphological analyses showed that the soma area was significantly larger in the microglia within the ipsilateral cortex of the stroke + clopidogrel-treated mice than in the stroke + control-treated mice (*p* = 0.0118) (Figure 6A). The MicroTrac software was also used to assess the microglia soma eccentricity, the number of primary branches, the total number of branch points, total branch length, cell area (µm^2^), cell solidity, and cell radius (µm), but no significant changes in these morphological parameters were detected for microglia within the ipsilateral cortex region between the stroke + control-treated and stroke + clopidogrel-treated mice.

Within the contralateral cortex, the morphological analyses revealed no significant changes in microglia number or morphology between the three groups (Figure 6B).

### 2.7. Short-Term Clopidogrel Administration Did Not Alter Neuronal Loss Post-Stroke

NeuN labeling, which is an established neuronal marker, was visualized to assess the number of neurons within the peri-infarct region (Figure 7A). Compared with the sham-operated mice, the number of NeuN-positive cells within the peri-infarct region was significantly reduced in both the stroke + control-treated mice (*p* = 0.0149) and stroke + clopidogrel-treaded mice (*p* = 0.0054). However, there was no difference in the number of NeuN-positive cells between the stroke + control-treated mice and stroke + clopidogrel-treaded mice (*p* = 0.1695) (Figure 7C).

### 2.8. Short-Term Clopidogrel Administration Did Not Alter Fibrinogen Post-Stroke

Traditionally, it was thought that bloodborne fibrinogen enters the subarachnoid space through a damaged BBB; however, a recent study showed that neurons are capable of producing fibrinogen chains [25]. Therefore, we also visualized fibrinogen labeling within the peri-infarct region as a secondary marker of neurons (Figure 7B). Consistent with the results for NeuN labeling, compared with the sham-operated mice, the number of fibrinogen-positive cells within the peri-infarct region was significantly reduced in both the stroke + control-treated mice (*p* = 0.0023) and stroke + clopidogrel-treaded mice (*p* = 0.0007), but there was no difference in the number of fibrinogen-positive cells within the peri-infarct region between the stroke + control-treated and stroke + clopidogrel-treated mice (*p* = 0.1654) (Figure 7D).

### 2.9. Short-Term Clopidogrel Administration Decreased the Number of T Cells Post-Stroke

CD3 labeling, which is an established T cell marker, was visualized to assess the T cell number within the ipsilateral and contralateral cortex regions (Figure 8A,B). Compared with the sham-operated mice, the number of CD3-positive cells within the ipsilateral cortex region was significantly elevated in both the stroke + control-treated mice (*p* < 0.0001) and stroke + clopidogrel-treated mice (*p* = 0.0439) (Figure 8C). Additionally, the number of CD3-positive cells in the ipsilateral cortex of the stroke + control-treated mice was significantly elevated beyond that of the stroke + clopidogrel-treated mice (*p* < 0.0001).

Within the contralateral cortex, there were no significant changes in the number of CD3-positive cells between the three groups (Figure 8D).

## 3. Discussion

Our study showed that clopidogrel negatively impacted cognitive performance when assessed using the PAL task. To our knowledge, this is the first demonstration of a significant impairment in learning and memory arising because of clopidogrel treatment. The learning and memory impairment observed in animals treated with clopidogrel is consistent with the concept that stroke increases vascular permeability, thus providing a pathway for clopidogrel to enter the brain and disrupt CNS repair processes. We further identified that clopidogrel increases the density of microglia but reduces the entry of CD3-positive cells. Interestingly, we found that neuronal cell loss was not greater in clopidogrel-treated animals relative to control-treated animals. Together, these findings suggest that clopidogrel and its metabolites may impair learning by influencing neuro-inflammatory-related processes post-stroke.

Cognitive impairment and dementia are known to occur at a rate of up to six times greater among stroke survivors than among the general population. There is currently little understanding of the mechanisms that lead to the development of dementia and cognitive impairment following stroke, although it appears that the pathophysiology is complicated and heterogeneous. Therefore, we sought to determine whether post-stroke clopidogrel administration was associated with reduced cognitive performance and/or reduced cognitive recovery post-stroke. Understanding the potential negative impacts of clopidogrel on cognitive recovery is challenging. First, these drugs are widely prescribed and are unquestionably effective for reducing secondary cardiovascular events, including stroke, which has led to their use as part of the ‘gold standard’ medical care. Second, dementia and cognitive impairment have long lead times, and thus, clinical trials aimed at detecting an increase in these outcomes would require large numbers of participants over extensive follow-up periods. To mitigate this, our approach capitalized on our team’s expertise in evaluating changes in cognition in mice using a touchscreen-based assessment of cognition, which is without question vastly superior to traditional assessment methodologies, such as the Morris water maze [26]. While traditional assessment methodologies have helped to advance our understanding, their relevance and relationship to the domains assessed in clinical studies were unclear and many assessed cognition while exposing the animals to significant levels of stress, thus potentially confounding the results. The touchscreen approach we utilized was low-stress and, more importantly, enabled an assessment of each mouse’s capacity for memory and learning, which are specific cognitive domains that are directly relevant to impairments described among human stroke survivors [24]. Looking at the results of our PAL task studies (Figure 1C), we observed that during the first week post-stroke, there was no difference in the correct response rate (associative memory) between the sham-operated, stroke + control-treated, and stroke + clopidogrel-treated mice. This is consistent with a previous study where there was no difference in learning and memory between sham-operated and stroke mice during the first three weeks of testing, with a significant difference only observed in the fourth week of testing [27]. By the end of week 2, we observed that the correct response rate had significantly improved for both the sham-operated and stroke + control-treated mice, which was expected and demonstrative of effective associative learning among the recovering animals. For the stroke + clopidogrel-treated mice, however, the correct response rate did not significantly improve relative to week 1, and the performance of the mice during week 2 remained significantly impaired compared with both the sham-operated and stroke + control-treated mice. These results provide evidence that the post-stroke administration of clopidogrel impairs learning and memory. Accordingly, we inferred that cerebrovascular leakage caused by the stroke allowed clopidogrel and its active metabolite to leak into the brain across the stroke-damaged BBB, where it may block microglia’s vital repair and learning and memory functions.

Previous studies showed that inducing cortical photothrombotic stroke in mice resulted in a clear impairment of motor function up to 3 months post-stroke, with the greatest motor deficits observed on day 28 post-stroke [28,29,30]. In our study, we also utilized a mouse photothrombotic stroke model that targeted the motor and somatosensory cortices. As expected, as early as post-stroke day 2, mice exhibited preferential use of their unaffected forelimb, as well as an increase in foot faults when using the impaired limb, indicating significant impairment of motor function. This impairment was consistent with previous studies [28,29,30] and was absent among sham-operated mice. Interestingly, whereas the degree of this motor impairment continued to increase among the stroke + control-treated mice progressing from post-stroke day 2 to day 7 and then day 14, the degree of motor impairment among the stroke + clopidogrel-treated mice was already maximal on day 2, with no further increase progressing to days 7 and 14 (Figure 2). The reason for this difference remains unclear but may indicate that clopidogrel administration hastened the development of neurological impairment. A previous study showed that neurological impairment was maximal on post-stroke day 28, with a slight improvement observed by day 84 [28]. Conducting a follow-up study of greater duration would, therefore, be valuable for determining whether motor impairment continues to worsen in the clopidogrel-treated mice to similarly reach a maximum on day 28, or whether maximal impairment occurs earlier (day 2) and then remains plateaued.

The risk of ischemic stroke ranges from 3 to 17% in the first three months after a minor ischemic stroke or a transient ischemic attack [31,32,33,34]. Several clinical trials showed that the risk of recurrent stroke decreased by approximately 20% following aspirin treatment [35,36]. The combination of aspirin plus clopidogrel treatment was shown to reduce the risk of stroke by 15% in patients with acute coronary syndromes and by 34% in patients who underwent percutaneous coronary intervention when compared with aspirin alone [37]. However, although treatment with aspirin plus clopidogrel in combination is reportedly more effective at preventing recurrent stroke, several trials showed that patients who received this combinational treatment had greater incidences of major and minor hemorrhage than those who received aspirin alone [38,39,40,41]. Additionally, although patients receiving the combinational treatment had a lower risk of ischemic stroke, myocardial infarction, or death from ischemic vascular causes, this benefit was restricted to the first month of the trial, whereas the risk of hemorrhage remained relatively constant throughout the three months of the trial [39]. Therefore, questions already existed as to the risk/benefit of post-stroke clopidogrel administration. If the cognitive impacts we observed in mice are found to also occur in humans, then this will have to be factored into future risk/benefit analyses. In addition to assessing the effect of clopidogrel on cognitive recovery, we also examined the impact of the short-term post-stroke clopidogrel administration on physiological parameters, namely, survival rate and body mass. Whilst 100% of the sham-operated mice (*n* = 27) and stroke + control-treated mice (*n* = 27) survived, we found that 3 out of the 26 clopidogrel-treated mice (11%) did not survive the 14 days of clopidogrel administration (Figure 3A). Whilst the precise reason these mice died is yet to be determined, extra- or intra-cranial hemorrhage following repeated clopidogrel administration is a feasible possibility in light of outcomes from the previously mentioned trials.

Our data showed that post-stroke, both the control-treated and the clopidogrel-treated mice experienced significantly greater weight loss than the sham-operated animals (Figure 3B). This is consistent with clinical observations from a phase IV clinical study of FDA data that showed that following clopidogrel (Plavix, generic) treatment, 2.5% of patients (3528 people) experienced unintentional weight loss. Over the 14-day study, however, the stroke + control-treated mice re-gained weight more rapidly than the stroke + clopidogrel-treated mice. Accordingly, by the end of the experiment (day 14), the body mass of the stroke + control-treated mice was not significantly different from that of the sham-operated mice, whereas the body mass of the stroke + clopidogrel-treated mice remained significantly reduced (Figure 3B). A study of greater duration is required to determine whether the clopidogrel-treated mice persist at significantly reduced body weights or improve to the level of sham-operated mice over time.

The BBB is a physical barrier that prevents the entry of blood components, toxic macromolecules, and drugs or substances used for drug-like effects, into the brain, therefore playing a role in maintaining the homeostasis of the CNS. The BBB is a dynamic structure composed of specialized endothelial cells, pericytes, astrocytes, and basement membranes, which each play a role in maintaining BBB integrity. Therefore, we aimed to determine whether the integrity of the BBB was compromised post-stroke. To investigate the BBB, we visualized the labeling of collagen IV, which is the most abundant component of the basement membranes and contributes to blood vessel integrity, stability, and functionality [42], and we observed a significant reduction in the number of collagen IV-positive cells within the peri-infarct region post-stroke (Figure 4A). This is consistent with previous studies, where decreased levels of collagen IV were observed in rat and mouse brains following ischemic stroke in a variety of ischemic stroke models [43,44,45,46]. To further ascertain whether the integrity of the BBB was compromised, we also visualized the labeling of IgG, which is the primary immunoglobulin found in blood and extracellular fluid. We detected significantly greater levels of IgG labeling within the peri-infarct region post-stroke, suggesting extravasation of IgG into the peri-infarct region across a compromised BBB (Figure 5). This is consistent with a prior study where vascular leakage was increased within the peri-infarct region post-stroke, as measured by IgG extravasation [47]. Together, these results provide evidence that a stroke compromised the integrity of the BBB within the peri-infarct region.

The active metabolite of clopidogrel under normal circumstances has low BBB permeability [48] and, as such, is not expected to enter the CNS, and thus, is not able to influence CNS cells, such as microglia. In testing this hypothesis, Lou et al. [21] showed that P2RY12-mediated activation of juxtavascular microglia promotes the rapid closure of small openings in the BBB, suggesting that microglia are among the first immune cells to respond to ischemic insults. The elimination of juxtavascular microglia via pharmacological inhibition or the genetic deletion of P2RY12 caused a delayed resealing of small openings in the BBB [21]. Moreover, it was shown that clopidogrel, in the absence of the vascular injury, did not suppress microglia process motility, whereas clopidogrel in the setting of vascular injury suppressed the movement of juxtavascular microglial processes toward photolytic lesions in the vessel wall, which, in turn, suppressed the closure of BBB leaks [21]. Interestingly, the non-P2RY12-dependent platelet antagonist, namely, acetylsalicylic acid, also known as aspirin, in the bloodstream in the setting of vascular injury did not reduce the motility of juxtavascular microglial processes toward injured capillaries [21]. Taken together, these data suggest that vascular injury can compromise the integrity of the BBB. Increased BBB permeability can, in turn, allow for direct access of drugs, such as clopidogrel, into the CNS, where it can alter the activities of juxtavascular microglia. Whilst we suggest that this mechanism provides one plausible explanation for how clopidogrel altered cognition in the current study, further experimental work would be required to verify this finding.

Our study was designed to model a clinically relevant scenario, where an individual would experience a stroke and then within 24 h would commence clopidogrel administration. Based on this scenario, we found similar results to Lou et al. [21] in that clopidogrel treatment exacerbated vascular leakage (Figure 5B); however, our microglia results were slightly different from those reported in the Lou et al.’s [21] study. They treated mice with clopidogrel prior to the injury, meaning that once there was a BBB breakdown, clopidogrel and its active metabolites were already present in the bloodstream, and therefore, able to enter the brain immediately, and thus, rapidly suppress the movement of juxtavascular microglial processes toward the injury. The microglial response to the stroke event is rapid; previous research showed that microglia appeared at the site of injury within 24 h, reached a peak at 4 to 7 days post-stroke, and then decreased by 14 days post-stroke [49]. As expected, we observed that photothrombotic stroke caused a significant increase in microglia number in the peri-infarct region (Figure 6A). Compared with the stroke + control-treated mice, we observed that after 14 days, the stroke + clopidogrel-treated mice had significantly more microglia present within the peri-infarct region, and those microglia exhibited altered morphology in the form of a significantly greater soma area (Figure 6A). Clinically, clopidogrel must be administered for several consecutive days to achieve a therapeutic effect, as the active metabolite of clopidogrel is produced in the liver [50]. Therefore, these results suggest that microglia were activated post-stroke and moved to the site of the injury; however, once clopidogrel crossed the stroke-damaged BBB, it inhibited further microglia movement. This manifested a significantly greater number of microglia within the peri-infarct region compared with the stroke + control-treated mice where the microglia remained able to move and vacate the peri-infarct region.

While our study identified a negative impact of clopidogrel on cognition, other investigators reported neuroprotective benefits of clopidogrel administration. Webster et al. [51] reported that clopidogrel-treated mice subjected to global cerebral ischemia suffered reduced neuronal injury, as measured by a neuron viability assay. Gelosa et al. [52] reported that rats subjected to middle cerebral artery occlusion and treated with ticagrelor, which is a reversible P2RY12 antagonist, had significantly diminished evolution of ischemic damage. Both studies credited the apparent neuroprotective effect of P2RY12 inhibition to the suppression of the inflammatory response to ischemic cellular injury. However, both studies were short (48–72 h), with only three injections of clopidogrel or ticagrelor administered. The duration of these two studies may not have been long enough for clopidogrel and/or ticagrelor to achieve functional inhibition of P2RY12 in vivo. It would be interesting to see whether these studies would report similar results if the duration of their experiments was longer. More recently, Dong et al. [53] assessed the effect of dual antiplatelet therapy with aspirin and clopidogrel on cognition in mice with transient distal middle cerebral artery occlusion. They used the Y-maze and novel object recognition tests for the assessment of cognitive function [53]. The results of the Y-maze test showed that short-term memory was restored only in the 12 mg/kg aspirin + clopidogrel group on day 21, but not on days 14 or 28 after ischemia [53]. The results of the novel object recognition test showed no significant difference in recognition memory at any of the time points for both doses of the aspirin + clopidogrel groups when compared with the control group [53]. Although valuable, both the Y-maze and novel object recognition tasks have several limitations; principally, these tasks share only limited similarities with the clinical assessment of cognitive function in stroke survivors.

A stroke triggers a cascade of events that lead to rapid neuron injury within the peri-infarct region. Initially, stressed neurons release damage-associated molecular patterns (DAMPs), which act as ‘eat-me’ signals, attracting microglial phagocytosis, and a prior study showed that microglia are capable of rapidly phagocytosing dead or dying neurons within hours post-stroke [54]. To assess neuron loss within the peri-infarct region, we quantitated NeuN labeling (Figure 7C), which is an established neuron marker; however, we also quantitated fibrinogen labeling (Figure 7D), as a recent study by Golanov et al. [25] showed that neurons are capable of producing fibrinogen chains. As expected, NeuN and fibrinogen labeling both indicated that the number of neurons within the peri-infarct region was significantly reduced in the stroke + control-treated mice, which is consistent with microglia having phagocytosed the damaged neurons. However, the neuron number was also significantly reduced in the stroke + clopidogrel-treated mice. We attributed this to the fact that clopidogrel administration did not commence until 24 h post-stroke, meaning microglia had sufficient time to migrate toward and phagocytose dead or dying neurons within the peri-infarct region before clopidogrel reached therapeutic levels that blocked P2RY12-mediated functions. As a follow-up, it would, therefore, be interesting to see what happens to neuron numbers in an experimental setting where clopidogrel and its active metabolites are already present in the bloodstream at the time of inducing stroke, thus allowing for rapid inhibition of microglial phagocytosis.

Following a stroke, the integrity of the BBB is compromised and cerebral microvessels become more permeable to molecules that are normally blocked from crossing the BBB [55]. All of these changes facilitate the immigration of systemic immune cells, including T cells, into the ischemic brain [55]. As expected, we observed a significant increase in the number of positive T cells within the peri-infarct region following photothrombotic stroke (Figure 8C). Interestingly, the clopidogrel treatment significantly reduced the number of T cells present within the peri-infarct region. This indicates that clopidogrel enters the brain across the stroke-damaged BBB, where it inhibits microglia activation and the subsequent microglial release of inflammatory mediators, thus inhibiting the recruitment of T cells into the peri-infarct region.

In summary, our data demonstrated that relatively short-term use of clopidogrel post-stroke negatively impaired cognition in mice. We observed this cognitive impairment using the PAL task. We were motivated to use the PAL task to assess cognition, as we previously observed that both mice and humans exhibit performance deficits on the PAL task after a stroke [24]. Although we did not definitely establish that the impairment in cognition was driven by the direct impact of clopidogrel on microglial-mediated repair processes, several markers that we considered are consistent with this hypothesis. Specifically, we confirmed that stroke was associated with greater levels of vascular permeability, which others showed is a pre-condition for the movement of clopidogrel and its metabolites into the CNS. We further observed that clopidogrel treatment altered microglial soma morphology and reduced the entry of CD3-positive cells in the peri-infarct area. Interestingly, we did not find changes in the neuron density, which is a finding that was reported in two other previous studies. While further work is clearly required, we believe that our results are important in highlighting the possibility of unanticipated cognitive impacts associated with current platelet-directed strategies for the prevention of secondary cardiovascular events that work through P2RY12 inhibition. If P2RY12 function does play a vital role in brain repair and cognitive recovery in humans, then the inadvertent ramifications of post-stroke clopidogrel administration could include significantly diminished cognitive recovery, which, in turn, may mean reduced patient independence, greater ongoing patient reliance on support services, and reduced quality of life. We could find very little in the way of literature aimed at examining the impact of drugs normally restricted to the periphery gaining access to the CNS after a stroke. We see this as a potentially new and important area for future work to consider.

## 4. Materials and Methods

### 4.1. Animals

All experiments were approved by the University of Newcastle Animal Care and Ethics Committee (A-2013-340) and conducted in accordance with the New South Wales Animals Research Act and the Australian Code of Practice for the use of animals for scientific purposes. This study was carried out using male C57BL/6 mice (ten weeks old), which were obtained from the Animal Services Unit at the University of Newcastle. Mice were maintained in a temperature- (21.0 ± 1.0 °C) and humidity-controlled environment with food and water available ad libitum. Lighting was used on a 12:12 h reverse light-dark cycle (lights on at 19:00 h), with all procedures conducted in the dark phase. Mice were acclimatized to the environment for 7 days before the initiation of the experiment. 

### 4.2. Experimental Design

Following the acclimatization period, mice were randomly allocated to one of the following three groups: sham, stroke + control, or stroke + clopidogrel. A total of 48 mice across two cohorts were used in this study. The first cohort of mice (sham, *n* = 9; stroke + control, *n* = 8; stroke + clopidogrel, *n* = 7) was assessed using a mouse touchscreen platform for the PAL task and underwent motor testing. The second cohort of mice (sham, *n* = 8; stroke + control, *n* = 8; stroke + clopidogrel, *n* = 8) was used for fixed tissue analysis (immunohistochemistry) (Figure 9).

### 4.3. Photothrombotic Occlusion

Photothrombotic occlusion was performed as previously described [24,27,28,29,30,47,56,57,58,59,60]. Briefly, on day 0 mice were anesthetized using 2% isoflurane and injected intraperitoneally with Rose bengal (0.2 mL of 10 mg/mL in 0.9% saline; Sigma-Aldrich, Sydney, NSW, Australia) for photothrombotic stroke or sterile 0.9% saline (0.2 mL; Pfizer, Sydney, NSW, Australia) for the sham procedure. After 8 min, the skull was illuminated for 15 min using a cold light source with a fiber optic end that was 4.5 mm in diameter placed 2.2 mm from the left lateral of Bregma onto the exposed skull. Occlusion was directed into the somatosensory and motor cortices, as these are common locations for stroke in humans (based on the Mouse Brain Atlas [61]). We used the photothrombotic occlusion model, as the model produces an interruption of blood flow caused by platelet aggregation and alterations of the BBB, which are consistent with stroke in humans.

### 4.4. Clopidogrel Administration

On day 1, animals commenced daily intraperitoneal injections of clopidogrel (40 mg/kg; Selleck Chemicals, Houston, TX, USA) or control (25% dimethyl sulfoxide (DMSO; Sigma, Sydney, NSW, Australia) in 0.9% saline).

### 4.5. Motor Testing

Motor tests were performed once pre-stroke (day 1) and 3 times post-stroke (days 2, 7, and 14). Forelimb asymmetry was assessed using a cylinder test, as previously described [28,29,30,59,60]. Briefly, each mouse was placed in a glass cylinder. The locomotor activity was recorded using video cameras from two different angles. The first forelimb to touch the wall of the cylinder during a full rear was scored as a wall placement. When both forelimbs (left and right) simultaneously touched the wall of the cylinder, it was considered as one placement for each forelimb. A total of 20 forelimb placements per mouse were scored by a blinded researcher. The forelimb asymmetry score was calculated as the ratio of the non-impaired forelimb placement minus the impaired forelimb placement to the total forelimb placement.

### 4.6. Cognitive Testing

Mouse touchscreen operant chambers (Campden Instruments Ltd., Loughborough, Leics, UK) were used in the cognitive testing, as previously described [24,27,28,47,58]. Briefly, mice were calorie-restricted overnight before the cognitive testing. A strawberry milkshake was used as a reward to motivate the performance of the mice. Mice underwent touchscreen training (habituation) before the experiment commenced. First, mice were habituated to the testing chambers for 20 min with a strawberry milkshake placed in the food tray. The mice then underwent initial touch training, where they learned to associate the delivery of the strawberry milkshake with the illumination of the reward tray every 30 s, with strawberry milkshake collection (drinking) triggering the start of the next trial. Additionally, visual stimuli (white squares) were displayed in each of the three response windows of the touchscreen, with touch responses producing the delivery of the strawberry milkshake. Next, during the must-touch training phase, the mice were required to associate a nose poke with a white square stimulus randomly displayed in one of three response windows of the touchscreen with the delivery of the reward. Touch responses on the other two response windows of the touchscreen had no programmed consequences. The mice had to complete 36 trials in 60 min during this training phase to proceed to the final phase. In the punish-incorrect training phase, the mice learned that touching the screen with no stimulus led to punishment with a 5 s activation of a house light, the absence of a reward, and a low tone. The mice had to perform the punish-incorrect training phase with a minimum correct rate of 75% for the training to be considered successful. The results of the training are illustrated in Figure 10. Following the training, the mice underwent photothrombotic occlusion surgery (day 0). On day 3 post-stroke, the mice commenced the PAL task. In the task, three stimuli images (flower, airplane, and spider) were associated with a specific spatial location (left, middle, right). In each trial, two images were displayed at the same time, one in the correct location and the other in an incorrect location. All trials were mouse-initiated and independent of the experimenter. If the mouse touched the image in its correct location, a tone was triggered, a reward was provided, and a correct trial was recorded. After the reward collection, the next trial was initiated. If the mouse touched the incorrect image or the correct image in its incorrect location, it was punished by the absence of a reward, a low tone, a 5 s activation of a house light, and an incorrect trial was recorded. An incorrect trial was followed by a correction trial in which the same configuration of stimuli was presented again until a correct response was made. Correction trials were not counted toward the total number of trials completed nor were they included in accuracy calculations. Each PAL session lasted for a maximum of 60 min or until the mouse completed 36 trials (excluding correction trials).

### 4.7. Perfusion, Tissue Collection, and Tissue Processing

On day 15, the animals were deeply anesthetized via intraperitoneal injection of sodium pentobarbital (0.2 mL of 325 mg/mL; Virbac, Milperra, NSW, Australia). The mice were perfused transcardially using ice-cold 0.9% saline followed by ice-cold 4% paraformaldehyde (PFA; Sigma-Aldrich, Sydney, NSW, Australia), pH 7.4. Brains were collected and post-fixed for 4 h in 4% PFA, pH 7.4, at 4 °C. Brains were then transferred to a 12.5% sucrose (Chem Supply, Gillman, SA, Australia) solution in 0.1 M phosphate-buffered saline (PBS; Thermo Fisher Scientific, Thornton, NSW, Australia) and stored at 4 °C until sliced. Brains were sliced (coronal sections) using a freezing microtome (Leica, Sydney, NSW, Australia) at a thickness of 30 μm, and then kept in an anti-freeze solution (0.05 M PBS, sucrose, and ethylene glycol) at 4 °C. Fixed brain sections were stored at 4 °C until used for histological analyses.

### 4.8. Immunohistochemistry

For immunoperoxidase labeling, free-floating 30 µm PFA-fixed sections were immunostained using standard protocols, as previously described [29,30,47,56,57,58,60,62]. Briefly, sections were washed with 0.1 M PBS and endogenous peroxidases were quenched in 0.1 M PBS containing 3% hydrogen peroxide (Sigma-Aldrich, Sydney, NSW, Australia). Non-specific binding was blocked with 3% normal horse serum (Sigma-Aldrich, Sydney, NSW, Australia) for 30 min. Sections were then incubated in the presence of antibodies against Collagen IV (cat# ab6586, Abcam, Cambridge, UK; 1:1000), Iba1 (cat#019-19741, WAKO, Richmond, VA, USA; 1:1000), NeuN (cat#MAB377, Millipore, Burlington, MA, USA; 1:500), Fibrinogen (cat#ab34269, Abcam, Cambridge, UK; 1:1000), or CD3 (cat#ab5690, Abcam, Cambridge, UK; 1:1000) with 2% normal horse serum for 72 h at 4 °C, washed 3 × 10 min with 0.1 M PBS, and then incubated in goat anti-rabbit IgG (cat#111-065-003; Jackson ImmunoResearch Laboratories, West Grove, PA, USA; 1:500) or goat anti-mouse IgG (cat#115-065-003; Jackson ImmunoResearch Laboratories, West Grove, PA, USA; 1:500) for 1 h at 25 °C. Following secondary antibody incubation, brain sections were washed for 3 × 10 min with 0.1 M PBS before the final incubation in 0.1% extravadin peroxidase (Sigma-Aldrich, Sydney, NSW, Australia) for 90 min. Brain sections were washed 3 × 10 min with 0.1 M PBS, and then immunolabeling was developed using a nickel-enhanced 3, 3′-diaminobenzidine (DAB; Sigma-Aldrich, Sydney, NSW, Australia) reaction. Tissues from all experimental groups were performed simultaneously and the DAB reactions were developed for the same length of time following the addition of glucose oxidase (Sigma-Aldrich, Sydney, NSW, Australia; 1:1000). After the processing was completed, sections were washed, mounted onto chrome alum-coated slides, and then cover-slipped.

### 4.9. Image Acquisition and Analyses

Images were taken at 40× with an Aperio AT2 (Leica, Wetzlar, Germany). For the IgG analysis, we calculated the percentage of cumulative threshold material for the range of pixel intensity values [63]. The pixel intensity level considered to be optimal for detecting genuine differences in immunoreactive signal was determined using ImageJ software (https://imagej.net/ij/index.html, accessed on 15 July 2023) to visualize the thresholding of cropped regions at individual pixel intensities. For the collagen IV analysis, we determined the percentage area covered by the labeling and the number of vessels using ImageJ software. For the NeuN, fibrinogen, and CD3 analyses, we determined the number of positive cells using automated ImageJ software. For Iba1 morphological analysis, we used the ‘MicroTrac’ program written in MATLAB [56,57,64,65,66,67]. This program is based on multilevel thresholding to identify cell soma, followed by the application of a minimum spanning tree algorithm to trace the cell processes. The MicroTrac program quantitates the number of cells per image, as well as various morphological parameters, including soma area (µm^2^), soma eccentricity, number of primary branches, the total number of branch points, total branch length, cell area (µm^2^), cell solidity (cell area divided by the convex hull), and cell radius (µm, calculated from the center of the cell to the end of the longest branch). These metrics were calculated for each cell in an image and averaged for each image.

### 4.10. Statistical Analyses

All data were analyzed using GraphPad Prism v9.1.0 (La Jolla, CA, USA) and are expressed as the mean ± standard error of the mean (SEM). Data were checked for normality using the Shapiro–Wilk test. For multiple comparisons, a one-way analysis of variance (ANOVA) followed by a two-stage Benjamini, Krieger, and Yekutieli test for controlling the false discovery rate was used. For the mixed-effects analysis, a two-way analysis of variance (ANOVA) followed by a two-stage Benjamini, Krieger, and Yekutieli test for controlling the false discovery rate was used. For the survival rates, a simple survival analysis (Kaplan–Meier) was performed. *p*-values ≤ 0.05 were considered statistically significant.

## Figures and Tables

**Figure 1 ijms-24-11706-f001:**
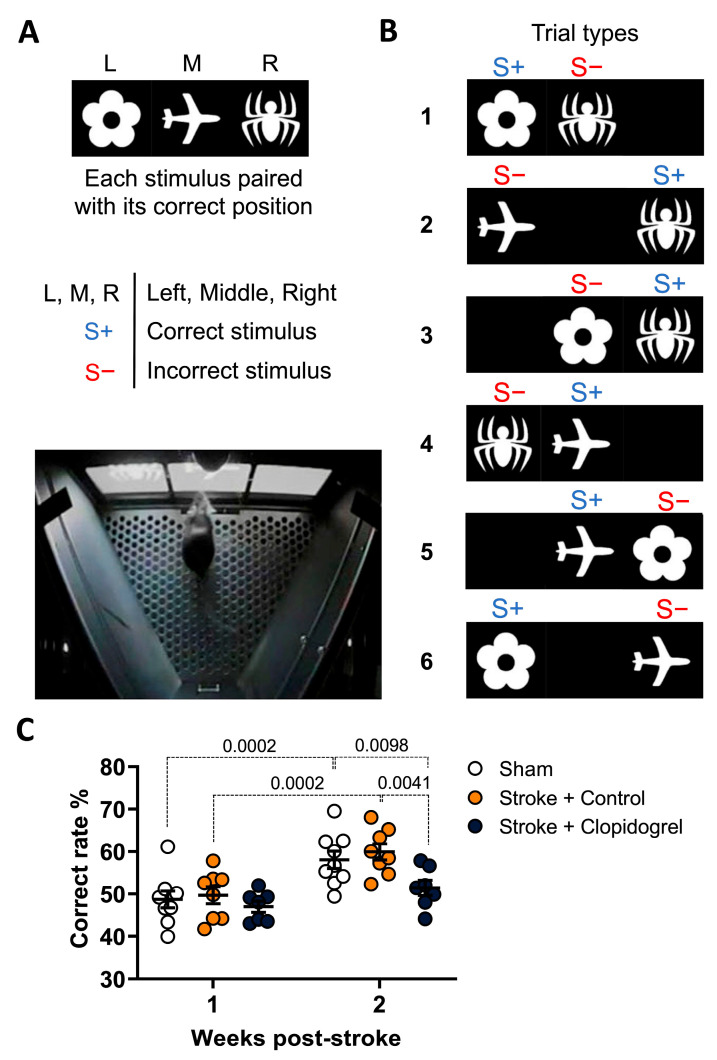
PAL task performance post-stroke. (**A**) A schematic depicting the stimuli used during the PAL task (flower, airplane, spider), which were only correct when displayed in the left (L), middle (M), or right (R) locations, respectively. (**B**) A schematic depicting six different types of trials (1–6) that occurred during the PAL task. Two different stimuli were presented in each trial: one was presented in its correct location (S+) while the other was presented in an incorrect location (S−). (**C**) The graph shows the percentages of correct responses of mice in each of the three groups (sham, stroke + control, stroke + clopidogrel) in the first and second weeks of treatment. Data were checked for normality using the Shapiro–Wilk normality test and then analyzed using two-way ANOVA with multiple comparisons (Benjamini, Krieger, and Yekutieli test). Data are given as the mean ± SEM. Significant *p*-values are indicated.

**Figure 2 ijms-24-11706-f002:**
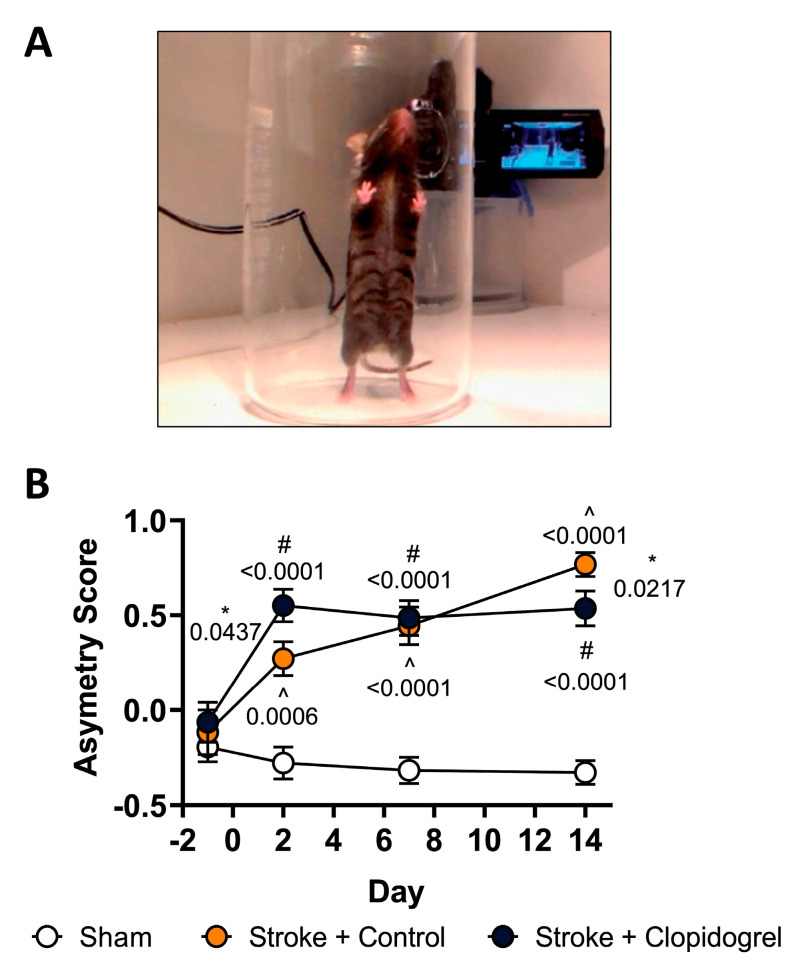
The effect of short-term clopidogrel administration on motor impairment. (**A**) Illustrates the placement of the mouse in the cylinder in which the spontaneous forelimb placement task was undertaken. (**B**) Cylinder task asymmetry scores for the three groups (sham, stroke + control, stroke + clopidogrel) at baseline (one day before stroke (/sham) surgery) and at 2, 7, and 14 days post-stroke. Data were checked for normality using the Shapiro–Wilk normality test and then analyzed using two-way ANOVA with multiple comparisons (Benjamini, Krieger, and Yekutieli test). Data are given as the mean ± SEM. Significant *p*-values are indicated. ^ denotes changes between the sham-operated and stroke + control animals, # denotes changes between the sham-operated and stroke + clopidogrel animals, and * denotes changes between the stroke + control and stroke + clopidogrel animals.

**Figure 3 ijms-24-11706-f003:**
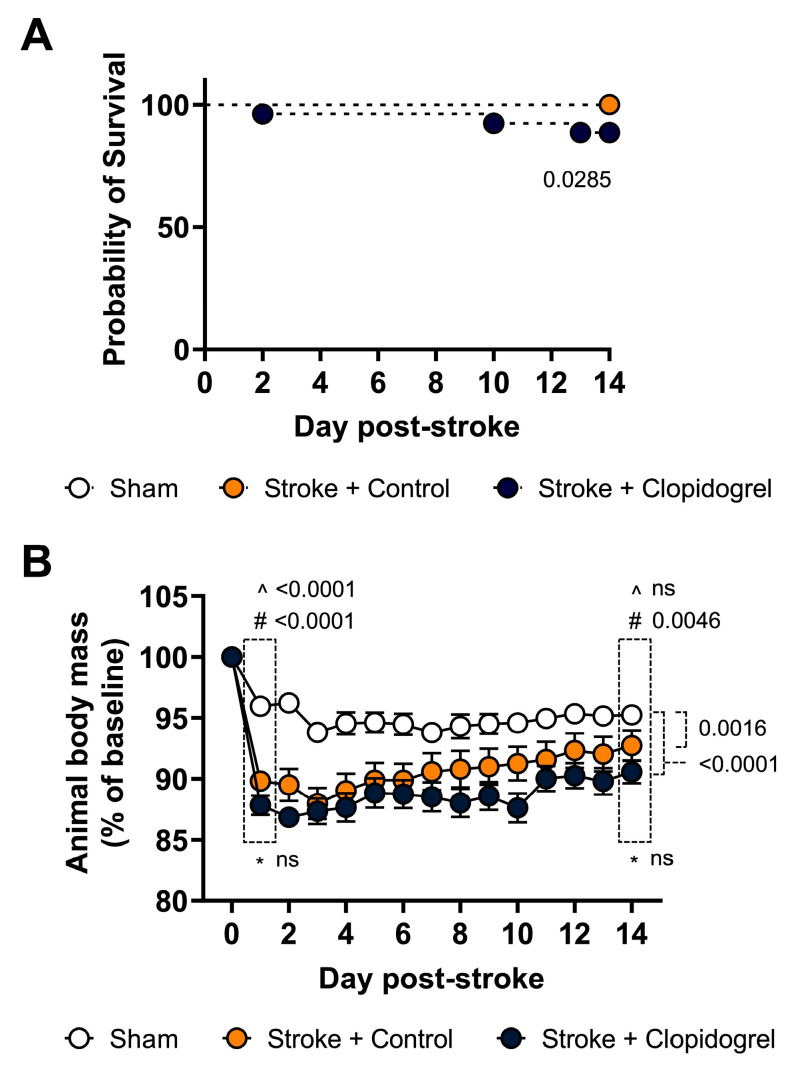
The effect of short-term clopidogrel administration on physiological parameters. (**A**) The survival rate and (**B**) body weight were recorded daily for 14 days post-stroke for the sham-operated mice, stroke + control-treated mice, and stroke + clopidogrel-treated mice. The body weight change was calculated as a percentage (%) change from day 0. The mouse survival rates were analyzed using a simple survival analysis (Kaplan–Meier). Mouse body weight data were checked for normality using the Shapiro–Wilk normality test and then analyzed using two-way ANOVA with multiple comparisons (Benjamini, Krieger, and Yekutieli test). Data are given as the mean ± SEM. Significant *p*-values are indicated. ^ denotes changes between the sham-operated and stroke + control animals, # denotes changes between the sham-operated and stroke + clopidogrel animals, and * denotes changes between the stroke + control and stroke + clopidogrel animals.

**Figure 4 ijms-24-11706-f004:**
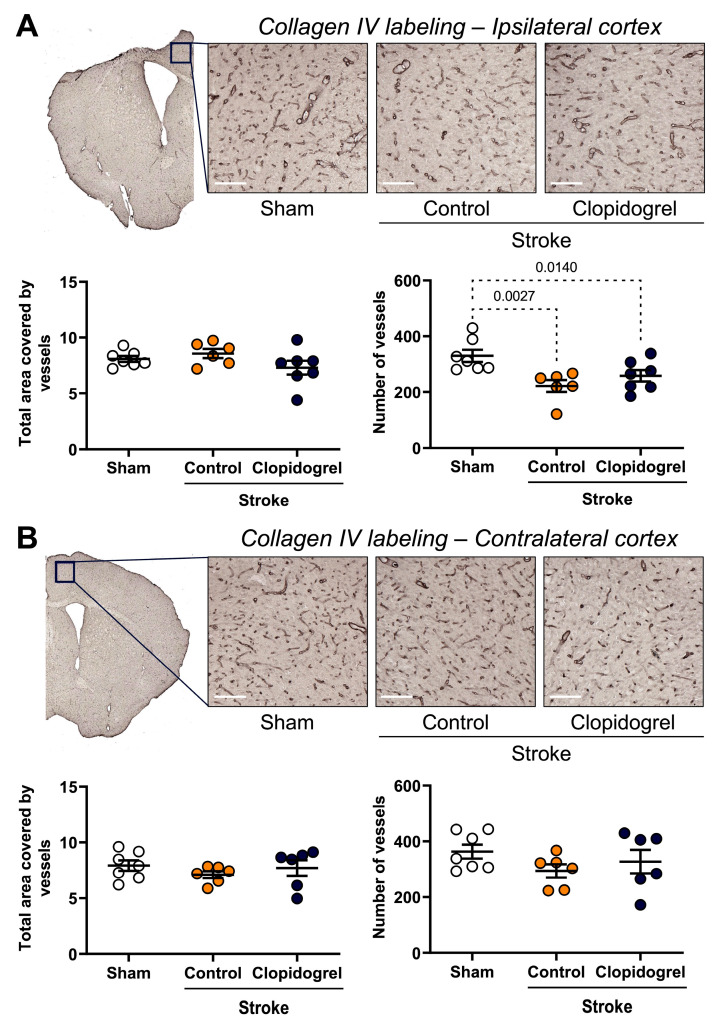
The effects of short-term clopidogrel treatment on vasculature within the ipsilateral and contralateral cortexes. Collagen IV labeling within (**A**) the ipsilateral cortex region and (**B**) the contralateral cortex region (scale bar = 80 µm). The top images show the regions examined and representative collagen IV labeling observed for the three groups: sham, stroke + control, and stroke + clopidogrel. The bar graphs underneath show changes in the total area covered by vessels and the number of vessels within the regions examined. Data were checked for normality using the Shapiro–Wilk normality test and then analyzed using two-way ANOVA with multiple comparisons (Benjamini, Krieger, and Yekutieli test). Data are given as the mean ± SEM. Significant *p*-values are indicated.

**Figure 5 ijms-24-11706-f005:**
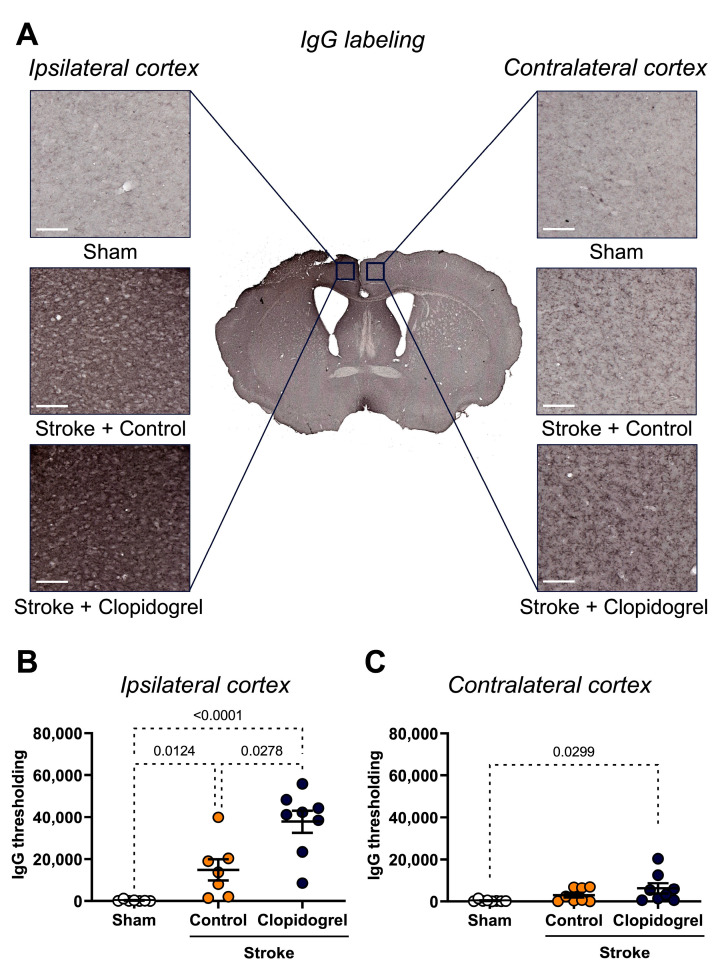
The effects of short-term clopidogrel treatment on vascular leakage in the ipsilateral and contralateral cortexes. (**A**) The location of the ipsilateral and contralateral cortex regions examined and representative labeling of IgG for the three groups: sham, stroke + control, and stroke + clopidogrel (scale bar = 60 µm). The bar graphs show the quantification of IgG thresholding within (**B**) the ipsilateral cortex and (**C**) the contralateral cortex. Data were checked for normality using the Shapiro–Wilk normality test and then analyzed using two-way ANOVA with multiple comparisons (Benjamini, Krieger, and Yekutieli test). Data are given as the mean ± SEM. Significant *p*-values are indicated.

**Figure 6 ijms-24-11706-f006:**
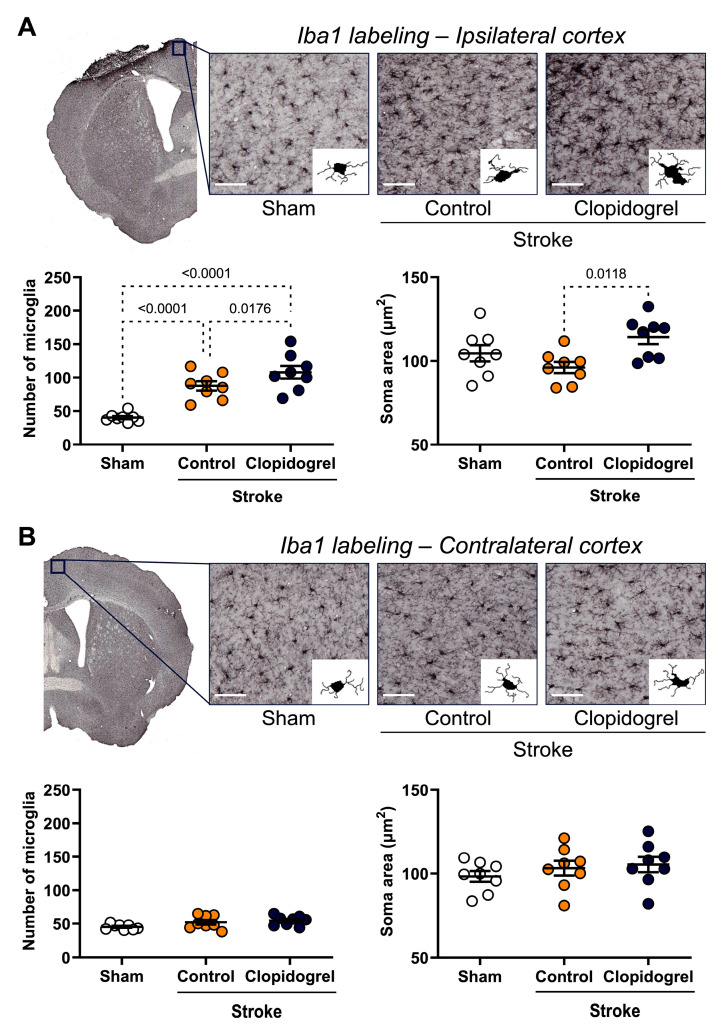
The effects of short-term clopidogrel treatment on the microglia number and morphology within the ipsilateral and contralateral cortexes. Iba1 labeling within (**A**) the ipsilateral cortex region and (**B**) the contralateral cortex region (scale bar = 60 µm). The top images show the regions examined and representative labeling observed for Iba1 for the three groups: sham, stroke + control, and stroke + clopidogrel. Microglia were reconstructed based on MicroTrac software detection of Iba1 labeling then, the microglia number and morphology were analyzed. The bar graphs underneath show the changes in the microglia number and soma area within the regions examined. All cells within one image were individually reconstructed and morphological parameters were averaged for each image, one per animal. Data were checked for normality using the Shapiro–Wilk normality test and then analyzed using two-way ANOVA with multiple comparisons (Benjamini, Krieger, and Yekutieli test). Data are given as the mean ± SEM. Significant *p*-values are indicated.

**Figure 7 ijms-24-11706-f007:**
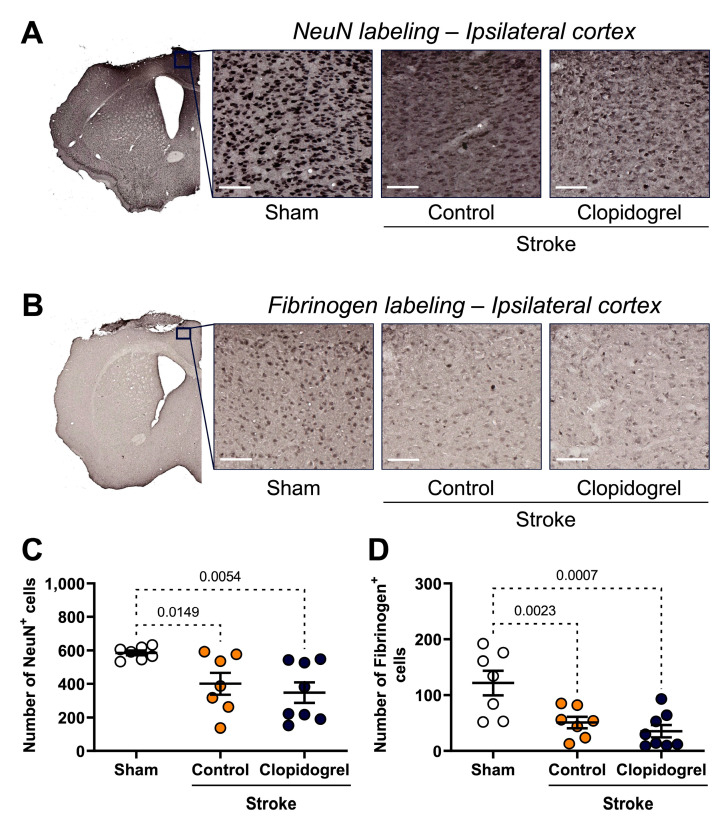
The effects of short-term clopidogrel treatment on neurons within the peri-infarct region. The location of the peri-infarct region examined and representative labeling of (**A**) NeuN-positive cells (scale bar = 65 µm) and (**B**) fibrinogen-positive cells (scale bar = 55 µm) for the three groups: sham, stroke + control, and stroke + clopidogrel. The bar graphs show the quantification of (**C**) the number of NeuN-positive cells and (**D**) the number of fibrinogen-positive cells. Data were checked for normality using the Shapiro–Wilk normality test and then analyzed using two-way ANOVA with multiple comparisons (Benjamini, Krieger, and Yekutieli test). Data are given as the mean ± SEM. Significant *p*-values are indicated.

**Figure 8 ijms-24-11706-f008:**
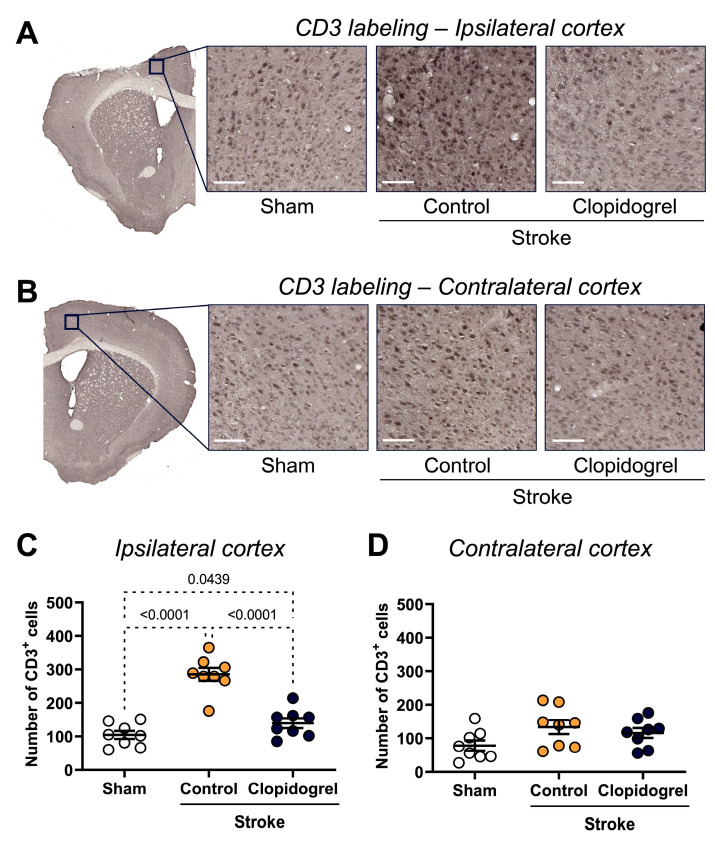
The effects of short-term clopidogrel treatment on the T cell number within the ipsilateral and contralateral cortexes. CD3 labeling within (**A**) the ipsilateral cortex region and (**B**) the contralateral cortex region (scale bar = 60 µm). Images show the regions examined and representative CD3 labeling observed for the three groups: sham, stroke + control, and stroke + clopidogrel. The bar graphs show the quantification of the CD3 labeling within (**C**) the ipsilateral cortex and (**D**) the contralateral cortex. Data were checked for normality using the Shapiro–Wilk normality test and then analyzed using two-way ANOVA with multiple comparisons (Benjamini, Krieger, and Yekutieli test). Data are given as the mean ± SEM. Significant *p*-values are indicated.

**Figure 9 ijms-24-11706-f009:**
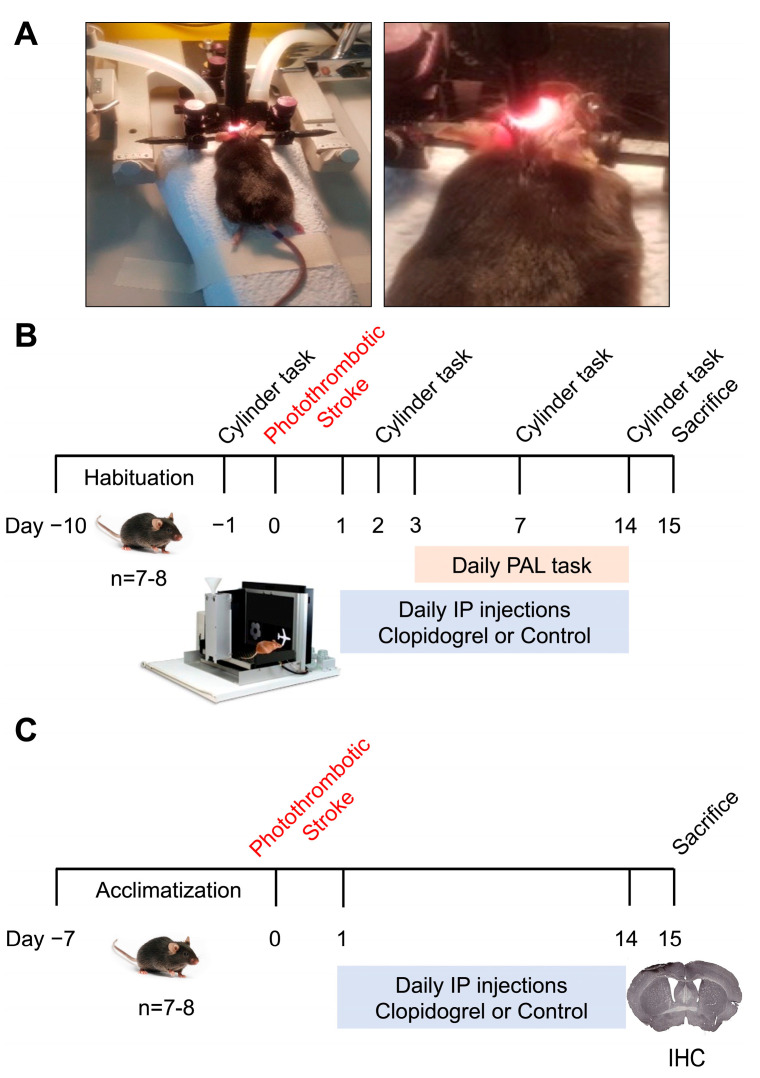
Experimental design timeline. (**A**) Photothrombotic occlusion set-up. (**B**) The first cohort of mice underwent motor impairment assessment using the cylinder task and memory and learning assessment using the PAL task. (**C**) The second cohort of mice was used for fixed tissue analysis (immunohistochemistry).

**Figure 10 ijms-24-11706-f010:**
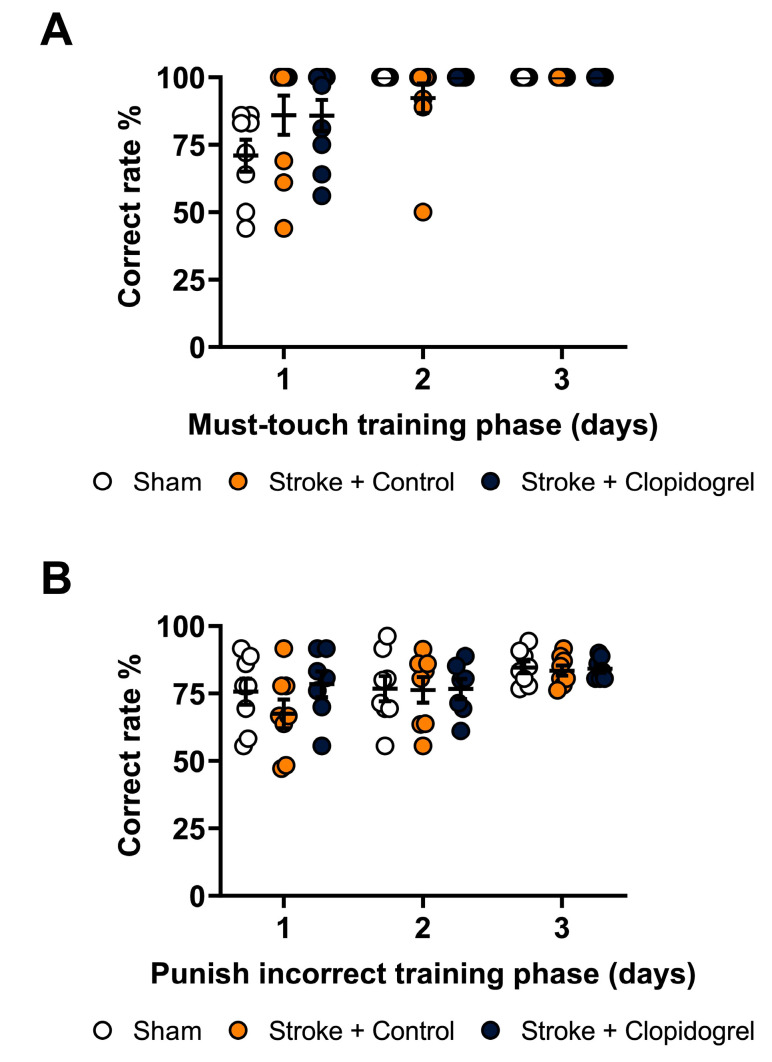
Touchscreen training. (**A**) Must-touch training phase. The mice had to complete 36 trials within 60 min during the must-touch training phase before proceeding to the next phase. (**B**) Punish-incorrect training phase. The mice had to attain a correct rate of 75% in the punish-incorrect training phase before commencing the PAL task.

## Data Availability

The data are contained within this article.

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
