# Peer review of "Clopidogrel Administration Impairs Post-Stroke Learning and Memory Recovery in Mice"

_ijms, 2023, doi:10.3390/ijms241411706_

Round 1
Reviewer 1 Report
The paper presented by Paul et al. evaluates the cognitive effects of the common use drug clopidogrel, a common use drug to prevent infarctions, including brain strokes, particularly used at the risk of recurrent strokes. The paper creates a directed brain stroke with the photoinduction of occlusion technique. The authors directed the infarction to the sesorial/motor cortices. They evaluate the consequences of clopidogrel use after the induced stroke, including an image recognition task in a touchscreen setting. The authors observed a deleterious effect of the drug in learning the task and also observed changes in the microglia activity and the number of neurons in the affected area. The authors conclude that changes in microglia activation and the BBB integrity are responsible for the cognitive deleterious effect of the drug.
The paper is properly written and explained. The data support the discussion and conclusions.
Authors should discuss how and why they choose the area to generate the stroke. Although the sensorimotor cortex is critical for many tasks and pretty well explains the motor deficit reported in the cylinder task, it is not the best strategy to evaluate cognitive damage because those structures are not implicated in the engrams that encode the memories evaluated in this paper. Logical areas for lesion evaluation would be the prefrontal cortex, parahippocampal cortices, and the hippocampus. Therefore, it is necessary to explain why such a target, how a stroke there would directly affect a memory task depending on other structures.
Maybe the authors wanted the stroke to not damage cognitive capabilities per se, only to allow the drug to generate cognitive impairment. If this is the chosen strategy, it would be necessary to have another group in which it is expected that the stroke per se would cause cognitive damage and the drug enhance this (or repair it). Although the authors explain that the drug causes major effects after a pathological disruption of the BBB, that is only an assumption, and there is no data on how clopidogrel affects cognitive and motor capabilities in mice not exposed to any stroke. Although unlikely by the reasoning presented by the authors, there is no way to discard that the cognitive impairment is stroke-independent. In any case, a deeper explanation of the experimental design would be critical to understanding every methodological decision directly affecting the results' interpretation.
Commonly when learning tasks take more than a session to acquire, researh groups report the result of one moment or probe test and the whole training process. As the training occurs after the stroke and during treatment, the learning curve would add important information on the cognitive process of the experimental groups.
Reviewer 2 Report
Clopidogrel is very frequently administered to prevent recurrence after acute stroke. The fact that this widely used medication might interfere with the repair processes that occur in the brain after acute stroke is very disturbing and requires clarification and potential implications in human stroke. The present study is very well designed, covering different aspects of the primary objective, from updated techniques for cognitive performance and motor function evaluation in the experimental animals subjected to photothrombotic stroke to immunohistochemistry analysis of the brain. The results are very clearly presented. I consider this an extremely valuable study which merits publication due the frequency with which clopidogrel is administered in patients to avoid recurrence of cardiovascular events. Nevertheless, in the discussion the authors should discuss the risk/benefit aspect related with the use of clopidogrel, which I consider at present to be in favor of continuing its use due to the evident beneficial effects on avoiding stroke recurrence. In the future, these very important results should impact in the search for other therapeutic agents.
The numerous studies conducted to date in humans with acute stroke have demonstrated its effectiveness in preventing stroke recurrence. Concerning its effect on cognition, the studies have not pointed toward an overt negative effect on cognition. For example, Kwan et al (2022) reported that antithrombotic therapy in individuals with small vessel disease on neuroimaging there was low-certainty evidence of no effect on cognitive outcomes as measured by the Cognitive Abilities Screening Instruments (CASI) assessed annually over five years. In a clinical trial to assess the tolerability, safety and intermediary pharmacological effects of cilostazol and isosorbide mononitrate, most of the individuals in the control group were receiving clopidogrel (Blair el al, 2019). The cognitive tests did not evidence significant differences between both groups.
Could the authors compare their results with those of Dong et al? Treatment with 12 mg/kg/day ASA + CPG improved spatial memory 21 days after ischemia in an acute minor stroke mouse model.
In general, the Discussion is very well written and addresses the main aspects of the study´s results; nevertheless, I suggest that the authors try to shorten it a bit. For example, the damage to the blood brain barrier secondary to stroke is repeated several times.
References:
Kwan J, Hafdi M, Chiang LLW, Myint PK, Wong LS, Quinn TJ. Antithrombotic therapy to prevent cognitive decline in people with small vessel disease on neuroimaging but without dementia. Cochrane Database Syst Rev. 2022 Jul 14;7(7):CD012269. doi: 10.1002/14651858.CD012269.pub2. PMID: 35833913; PMCID: PMC9281623.
Dong W, Liu X, Liu W, Wang C, Zhao S, Wen S, Gong T, Chen W, Chen Q, Ye W, Li Z, Wang Y. Dual antiplatelet therapy improves functional recovery and inhibits inflammation after cerebral ischemia/reperfusion injury. Ann Transl Med. 2022 Mar;10(6):283. doi: 10.21037/atm-22-735. PMID: 35433995; PMCID: PMC9011245.
Blair GW, Appleton JP, Flaherty K, Doubal F, Sprigg N, Dooley R, Richardson
C, Hamilton I, Law ZK, Shi Y, et al. Tolerability, safety and intermediary
pharmacological effects of cilostazol and isosorbide mononitrate, alone
and combined, in patients with lacunar ischaemic stroke: the LACunar
Intervention-1 (LACI-1) trial, a randomised clinical trial. EclinicalMedicine.
2019;11:34–43. doi: 10.1016/j.eclinm.2019.04.001
